# Determinants of the Level of Anti-SARS-CoV-2 IgG ANTibodiEs after Vaccination (DANTE-SIRIO 7) Study in a Large Cohort of Healthcare Workers

**DOI:** 10.3390/vaccines10122125

**Published:** 2022-12-12

**Authors:** Magdalena Krintus, Maciej Piasecki, Piotr Lackowski, Katarzyna Buszko, Aldona Kubica, Agata Kosobucka-Ozdoba, Piotr Michalski, Lukasz Pietrzykowski, Wioleta Stolarek, Agata Wojcik, Maria Tomczak, Emilia Wojtal, Jacek Krys, Zbigniew Wlodarczyk, Jacek Kubica

**Affiliations:** 1Department of Laboratory Medicine, Collegium Medicum in Bydgoszcz, Antoni Jurasz University Hospital No. 1 in Bydgoszcz, Nicolaus Copernicus University in Torun, 85-094 Bydgoszcz, Poland; 2Department of Cardiology and Internal Medicine, Collegium Medicum in Bydgoszcz, Antoni Jurasz University Hospital No. 1 in Bydgoszcz, Nicolaus Copernicus University in Torun, 85-094 Bydgoszcz, Poland; 3Department of Theoretical Foundations of Biomedical Science and Medical Informatics, Collegium Medicum in Bydgoszcz, Nicolaus Copernicus University in Torun, 85-094 Bydgoszcz, Poland; 4Department of Health Promotion, Collegium Medicum in Bydgoszcz, Antoni Jurasz University Hospital No. 1 in Bydgoszcz, Nicolaus Copernicus University in Torun, 85-094 Bydgoszcz, Poland; 5Department of Pharmacology and Therapeutics, Collegium Medicum in Bydgoszcz, Antoni Jurasz University Hospital No. 1 in Bydgoszcz, Nicolaus Copernicus University in Torun, 85-094 Bydgoszcz, Poland; 6Department of Exercise Physiology and Functional Anatomy, Collegium Medicum in Bydgoszcz, Nicolaus Copernicus University in Torun, 85-094 Bydgoszcz, Poland; 7Department of Transplantology and General Surgery, Collegium Medicum in Bydgoszcz, Antoni Jurasz University Hospital No. 1 in Bydgoszcz, Nicolaus Copernicus University in Torun, 85-094 Bydgoszcz, Poland; 8Department of Public Health, Collegium Medicum in Bydgoszcz, Antoni Jurasz University Hospital No. 1 in Bydgoszcz, Nicolaus Copernicus University in Torun, 85-094 Bydgoszcz, Poland

**Keywords:** healthworkers, anti-SARS antibodies, vaccination, immunity, adverse reactions

## Abstract

The aim of this study was to determine anti-SARS-CoV-2 IgG concentrations and their major determinants in healthcare workers (HCWs) after full vaccination with the BNT162b2 vaccine. We recruited 847 individuals vaccinated with two doses of the BNT162b2 vaccine, who completed the questionnaire, and whose antibody concentrations were tested after 3 and 6 months after full vaccination. Anti-SARS-CoV-2 IgG levels were measured on the routinely employed Siemens Atellica system. The cutoff for positivity was ≥21.8 BAU/mL. Three and 6 months after vaccination, the majority of participants were seropositive. Median concentrations of anti-SARS-CoV-2 IgG significantly decreased from 1145 BAU/mL (IQR: 543–2095) to 225 BAU/mL (IQR: 100–510). Major positive determinants of antibody levels were fever after both doses of vaccine, prior-COVID-19 exposure, and muscle pain after the first dose. Lack of symptoms after the second dose and time since vaccination were significant negative determinants of anti-SARS-CoV-2 IgG concentrations. No other factors, including age and gender, or underlying comorbidities had a significant effect on antibody levels in HCWs. The anti-SARS-CoV-2 response after two doses of BNT162b2 vaccine was independently associated with prior-COVID-19 exposure, time since vaccination, and the occurrence of symptoms after either dose of vaccine. Easily reportable adverse reactions may facilitate the identification of immune response in HCWs.

## 1. Introduction

Since the European Medicines Agency (EMA) granted a first, conditional authorization for messenger RNA (mRNA)-based BNT162b2 vaccine COMIRNATY (Pfizer/BioNTech, Reinbek, Germany), nearly 108 million of doses have been distributed to Poland with a still lower cumulative uptake of fully vaccinated Polish adults of 59.4% when compared to 73.3% fully vaccinated European Union (EU) citizens. Although this global and unprecedented vaccination campaign against COVID-19 led to a significant decrease in symptomatic disease, deaths, and hospitalization worldwide, 37.1% of population remains not fully vaccinated [1].

COVID-19 vaccines are designated to elicit high levels of neutralizing antibodies that target the spike’s S1 protein subunit and receptor-binding domain (RBD) [2]. BNT162b2 vaccine is administered as an intramuscular injection in a primary series of two doses 21 days apart. A person is considered fully vaccinated against SARS-CoV-2 infection ≥ 2 weeks after receipt of the second dose in a two-dose series. Encouraging data from two phase III BNT162b2 vaccine efficacy trials and randomized clinical trials demonstrated up to 95% efficacy following a two-dose vaccination regimen [3,4]. Nevertheless, additional studies suggested that the effectiveness of mRNA vaccines against hospitalization and severe course of disease wanes after about 4 months [5,6]. Even though antibody testing is currently not recommended to assess the need for vaccination in unvaccinated individuals or to assess immunity to SARS-CoV-2 following vaccination [7], vaccine-induced antibody development has great implications for antibody testing. Commercially available, routine serological methods for antibodies measurement are fast, easily accessible, and relatively inexpensive tools to evaluate the humoral immunity following vaccination. Moreover, their role and importance for public health in monitoring and responding to the COVID-19 pandemic, and their clinical utility in delivering patients care have increased significantly. Although, due to antibody testing, the understanding of the humoral immune response to vaccination against SARS-CoV-2 is rapidly advancing, the robustness and durability of vaccine-induced immunity and how it compares with prior-COVID-19 infection, as well as the severity of symptoms following each dose of vaccine, remain unknown. As such, immune response to vaccination represented by antibody concentrations may be affected by many both positive and negative factors which have to be carefully researched and identified. Promoting vaccination strategies and understanding the mechanisms of vaccine immunity seem to be priorities in response to the constantly changing epidemic situation in the world and the emergence of new SARS-CoV-2 variants with a still unsatisfactory level of global vaccination.

Therefore, our study was aimed at evaluating anti-SARS-CoV-2 IgG concentrations, after two doses (the full vaccination cycle) with the BNT162b2 vaccine at 3 and 6 months post vaccination. We compared these results with the COVID-19 history and severity of symptoms during the disease and following vaccination to correlate major determinants of antibody response after vaccination.

## 2. Material and Methods

### 2.1. Study Design and Participants

The study was designed as a prospective, observational, single-center study (DANTE-SIRIO 7) in healthy, unselected adult volunteers recruited from the staff of Dr. Antoni Jurasz University Hospital No.1 in Bydgoszcz (Poland), which is the region’s largest high-specialty medical facility, consisting of 26 departments and units (summarily 914 beds), eight diagnostic divisions, and 51 specialist outpatient clinics. The hospital employs around 2000 healthcare workers (HCWs) and annually admits approximately 39,800 patients and provides approximately 200,000 specialist consultations. HCWs comprised a study cohort which included various occupational groups: clinicians, nurses, laboratory and other medical professionals, and technical and administrative hospital staff.

Initially, 1000 adult participants were planned to be included in the study. Finally, we recruited 847 unselected individuals who met the inclusion criteria, i.e., who were vaccinated with two doses of the BNT162b2 vaccine, administrated 21 days apart, who completed the sociodemographic, anthropometric, and clinical questionnaire, and whose serum samples for antibody concentration testing were collected 3 months (92 ± 3 days) and 6 months (179 ± 5 days) after administration of the second vaccine dose. Vaccination data were verified in the hospital vaccination register center. Subjects with prior-COVID-19 were defined as having RT-PCR-confirmed exposure to SARS-CoV-2 any time before vaccination. Participants were recruited during the peaks of the second and third waves of the pandemic in Poland, and sampling for anti-SARS-CoV-2 IgG measurements was performed between April 2021 and November 2021 at 3 months post vaccination, and between July 2021 and January 2022 at 6 months post vaccination. A detailed study protocol was previously published elsewhere [8]. A flowchart for selection of the study participants is shown in Figure 1.

The study protocol was approved by the Ethics Committee of Nicolaus Copernicus University in Torun, Ludwik Rydygier Collegium Medicum in Bydgoszcz (KB 160/2021), and all participants signed written informed consent prior to enrollment in the study. The DANTE-SIRIO 7 study was registered in the ClinicalTrials.gov database (NCT05109585), (accessed 20 September 2022).

### 2.2. Questionnaire

At the first timepoint (3 months after vaccination), participants completed a detailed clinical questionnaire on sociodemographic and anthropometric characteristics, comorbidities, history of previous SARS-CoV-2 infection, and accompanying adverse reactions (ARs) occurring after each dose of vaccination (16 questions in total) (Appendix A). At the same time, by means of an additional survey, possible SARS-CoV-2 infection was reported in each phase of the study.

### 2.3. SARS-CoV-2 IgG Antibody Testing

Fresh serum samples were used to measure anti-SARS-CoV-2 IgG concentrations against the spike protein RBD on the routinely employed Siemens Atellica system (Siemens Healthineers, Erlangen, Germany). For each participant, serum sample collection and antibody measurements were performed 3 and 6 months after the second dose of BNT162b2 according to the manufacturer’s procedures.

The Atellica IM sCOVG assay is a fully automated two-step sandwich immunoassay with acridinium-ester chemiluminescent technology. Results of SARS-CoV-2 IgG were initially given as U/mL; however, to achieve assay traceability to WHO 20/136 (BAU/mL), all reported results were multiplied by 21.8, whereby the cutoff for positivity was defined as ≥1.0 U/mL (21.8 BAU/mL). The manufacturer reports a range of quantification of 0.5 to 150.0 U/mL, which may be extended to 750.0 U/mL upon automated 1:5 predilution. The lower limit of the analytical measuring interval corresponded to 0.50 U/mL. The assay has a specificity of 99.90% (95% CI 99.64–99.99) and a sensitivity of 96.41% (95% CI 92.74–98.44) for samples ≥ 21 days post RT-PCR confirmation.

### 2.4. Statistical Methods and Data Evaluation

Quantitative data are shown as medians and interquartile ranges (IQRs), while categorical data are shown as numbers and percentages. The Shapiro–Wilk test was used to control distribution. Due to a non-Gaussian distribution, the Mann–Whitney U-test was applied to evaluate differences between two continuous variables. Categorical variables within two groups were compared using the chi-square 2 × 2 test or Fisher’s exact test for independence. The linear mixed model (LMM) was applied to evaluate the association between anti-SARS-CoV2-IgG as the dependent variable and age, gender, BMI, sample timing, history of prior-COVID-19 infection, underlying comorbidities, and ARs after each dose of vaccine as independent variables. The obtained results were reported as the average changes in anti-SARS-CoV2-IgG concentrations in BAU/mL with standard error (SE) and coefficient of determination (R^2^). A *p*-value less than 0.05 was considered statistically significant. Statistical analysis was performed using R software 4.0.4 (R foundation, Vienna, Austria) and PQStat 1.8.4 (PQStat Software, Poznan, Poland).

## 3. Results

### 3.1. Characteristics of Participants

Baseline characteristics of study participants (*n* = 847) with relation to a previous COVID-19 infection are presented in Table 1.

The study participants were predominantly women (79.9%), with a median age of 45 years (IQR: 34–53). Around 30% of participants were overweight, 15% were obese, and 49% had underlying comorbidities. Individuals with comorbidities were older with a median age of 47 (IQR: 38–57) years when compared to those who did not report any underlying disease [median age 40 (29–49) years]. A total of 246 participants had been infected with SARS-CoV-2 before vaccination. Most of them were subjected to home isolation, while only 12 (4.9%) required hospitalization. The frequency of prior COVID-19 infection was significantly higher in patients with higher BMI, hypertension, and cardiac disease. Eleven individuals (1.3%) of those fully vaccinated had vaccine breakthrough infection within 6 months (173 ± 47 days) after the second dose of BNT162b2. The emergence of a highly transmissible and more pathogenic delta variant (B.1.617.2) of SARS-CoV-2 in Poland [9] seemed to be responsible for breakthrough infections observed in our study.

### 3.2. Characteristics of Post-Vaccine Adverse Effects after the First and the Second Dose of Vaccine

Although the majority of participants suffered at least one post-vaccine AR after either dose, almost every fifth participant did not experience any post-vaccination symptoms regardless of the dose of vaccine. The most frequently reported AR was injection site soreness, followed by muscle pain and malaise (Table 2). There were statistically significant differences between frequencies of systemic ARs after first and second dose of vaccine. Higher frequencies of reactogenicity in terms of malaise, feverish state and fever, muscle pain, gastrointestinal complaints, headache, and other unspecific symptoms were observed after the second dose of BNT162b2 vaccine. Contrarily, all systemic ARs were less frequent after the first dose of vaccine.

### 3.3. Anti-SARS-CoV-2 IgG Concentrations 3 and 6 Months after Vaccination

Furthermore, 90 ± 3 days after full vaccination cycle with the BNT162b2, the majority of participants were seropositive (99.4%); similarly, 270 ± 3 days after vaccination, 99.6% remained seropositive. Median concentrations of anti-SARS-CoV-2 antibodies significantly decreased from 1145 BAU/mL (IQR: 543–2095) to 225 BAU/mL (IQR: 100–510) (Table 1, Figure 2). The decrease in antibody levels was irrespective of previous exposure to SARS-CoV-2, but its magnitude was greater for those without prior evidence of COVID-19 infection. Surprisingly, we noted no significant difference in antibody concentrations between individuals with and without underlying comorbidities [1195 (516–2213) vs. 1060 (547–1964) *p* = 0.245, respectively] measured after 3 month follow-up and, similarly, 6 months after vaccination [257 (130–619) vs. 240 (134–515) *p* = 0.376, respectively].

### 3.4. Association of Prior-COVID-19 Infection with Anti-SARS-CoV-2 IgG Concentrations after Vaccination

Those exposed previously to SARS-CoV-2 had 2.5-fold and threefold higher median antibody levels when compared to naïve individuals 3 and 6 months post vaccination, respectively. Interestingly, participants who had breakthrough infections did not differ significantly from those with prior-COVID-19 with regard to age, BMI, and anti-SARS-CoV-2 IgG levels at the 3 and 6 month follow-ups.

### 3.5. Determinants of Anti-SARS-CoV-2 IgG Levels Post Vaccination

We used mixed linear regression models with random effects to estimate individual-specific determinants of anti-SARS-CoV-2 IgG levels measured 3 and 6 months after vaccination with two doses of BNT162b2. We first fitted a univariable logistic models evaluating the impact of independent variables on anti-SARS-CoV-2 IgG concentrations. We found that this model identified a number of independent variables as being statistically significant positive or negative determinants of antibody response.

In the course of further analysis, we fitted a multivariable model which included statistically significant parameters retrieved from univariable regression. This model identified six determinants which remained significantly associated with antibody levels after vaccination (Appendix A). Figure 3A–D graphically depict antibody concentrations with regard to prior-COVID-19 infection, as well as lack of symptoms and fever after the first and the second dose of vaccine 3 and 6 months post vaccination, respectively.

To quantify the impact of determinants on SARS-CoV-2 IgG concentrations, we derived the average absolute change in antibody levels expressed in BAU/mL for each determinant found to be statistically significant in the univariable and multivariable models (Figure 4A,B). In both univariable and multivariable analyses, the occurrence of fever after taking the first dose of vaccine and prior exposure to SARS-CoV-2 were the most potent positive determinants of immune response quantified by antibody levels, while time elapsed since the completion of vaccination course and lack of symptoms after the second dose of vaccine were significant negative predictors of antibody concentrations. The magnitude of increase in SARS-CoV-2 IgG levels ranged from 1775 to 418 BAU/mL for all significant positive determinants in univariable analysis, and from 1046 to 297 BAU/mL for all significant positive determinants in multivariable analysis. The time since vaccination at the 3 and 6 months follow-up was responsible for an average decrease in antibody levels by 1000 BAU/mL, whereas lack of symptoms after the second dose of vaccine caused a significant decrease in SARS-CoV-2 IgG levels by 331 and 280 BAU/mL in univariable and multivariable modeling, respectively.

Lastly, we developed models with interaction which revealed statistically significant interactions between independent variables. Time since vaccination interacted significantly with prior-COVID-19 exposure, as well as fever after first and second dose, whilst positive COVID-19 history was interrelated to the occurrence of fever after first and second dose of the vaccine.

## 4. Discussion

Better understanding of the relationship between immune response and protection against SARS-CoV-2 infection obtained with vaccination is still (and urgently) needed to optimize vaccination strategies in order to generate the highest possible immunity of global population. In our study, we investigated a large cohort of HCWs, a specific group clinically vulnerable to SARS-CoV-2 exposure, but also well vaccinated, at 3 and 6 months follow-up after completing full vaccination cycle with BNT162b2 vaccine. Our observations indicated a number of significant determinants of the immune response measured and correlated with anti-SARS-CoV-2 IgG concentrations. Among them, the highest post-vaccination immunity, as quantified by an average increase in antibody levels, was associated with the occurrence of fever after both first and second dose of vaccine, as well as prior-COVID-19 infection. Muscle pain after administration of the first dose of BNT162b2 had a lower but still significant impact on the antibody response. In contrast, lack of symptoms after second dose of the vaccine and time since vaccination were significant negative determinants of anti-SARS-CoV-2 IgG concentrations. Surprisingly, no other factors, including age and gender, or underlying comorbidities had a significant and measurable effect on antibody levels in this large cohort of HCWs.

Although there is a large evidence base on the effectiveness of mRNA vaccines against COVID-19, little is still known about factors which may influence protection, including adverse reactions and how they correlate with vaccine-induced immunogenicity. In clinical trials, the most common post-vaccination symptoms after taking BNT162b2 were mild and local Ars, e.g., injection site pain, redness, and swelling, whilst common mild to moderate systemic ARs were fatigue, headache, fever, and muscle pain [3,4,10,11,12]. Similarly, in our study the most common reported AR were injection site soreness, muscle pain, malaise, headache, feverish state, and fever. Local and systemic reactogenicity occurred more frequently after the second dose of vaccine than after the first [13,14,15]. The intensity and frequency of ARs were clearly dose-dependent, and the second dose of vaccine was a strong stimulator of systemic but not local ARs. Findings of a large online cohort study suggested that some individuals may experience more adverse effects following vaccination but severe ARs are rare. This study identified full vaccination cycle, vaccine brand, younger age, female sex, and having had COVID-19 before vaccination as being associated with greater odds of adverse effects [12]. This may explain a relatively high frequency of ARs in our HCW cohort as it comprised predominantly middle-aged women, fully vaccinated with BNT162b2, of whom almost one-third were previously exposed to SARS-CoV-2. Surprisingly, not all reported systemic ARs were correlated with antibody concentrations. In the multivariable analysis, the strongest AR-related determinants of immune response quantified by anti-SARS-CoV-2 IgG concentrations were fever after each dose of vaccine and muscle pain after first dose. In contrast, a lack of any adverse symptoms after the second dose of BNT162b2 was associated with a significant decrease in antibody levels 6 months post vaccination. Due to waning antibody levels over time, an observation interval of 3 and 6 months post vaccination seems to be particularly important. Recent studies revealed that the effectiveness of mRNA vaccines against hospitalization and severe course of disease wanes and correlates with decreased anti-SARS-CoV-2 antibodies a few months after vaccination [6,16,17,18,19,20]. In our study, time since vaccination, was the strongest negative predictor of waning immunogenicity expressed by antibody levels against spike protein RBD measured at 3 and 6 months post vaccination. The magnitude of waning antibody levels over follow-up time was more than threefold higher than the decline in antibody concentrations observed in individuals without symptoms after the second dose. Nevertheless, the immune response quantified by anti-SARS-CoV-2 IgG concentrations remained robust, and the seroprevalence was high in the entire cohort of HCWs throughout the study period. It seems, therefore, plausible to identify factors that could improve the humoral immune response and, consequently, optimize the protection against SARS-CoV-2 infection. Our study identified fever after first dose of vaccine, prior-COVID-19 infection, fever after the second dose, and muscle pain after the first dose as the strongest and significant predictors of elevated anti-SARS-CoV-2 concentrations after adjustment for other factors. Although the precise functional effects of human body temperature fluctuations are still largely unknown, substantial scientific evidence suggests that febrile temperatures increase the effectiveness of the immune response to infections by activating both the innate and the adaptive immune systems [21,22,23]. Nevertheless, there are conflicting findings on the association between fever and post-vaccination immunogenicity. While some studies showed no relationship between the presence or lack of individual post-vaccination response including fever and the generation of SARS-CoV-2 spike-specific IgG or neutralizing antibodies caused by vaccination [24,25], others are in line with our observations [15,26,27], suggesting that fever may significantly trigger the immune response expressed by higher antibody levels against SARS-CoV-2 spike protein. Different statistical approaches (adjustment for other confounding factors in multivariable modeling), lack of information about anti-inflammatory medication used before and after vaccination, and differences in the characteristics in investigated populations may be responsible for conflicting results in the literature. Taken together, the impact of easily reportable ARs on anti-SARS-CoV2 antibody levels may be underestimated.

Prior-COVID-19 infection is a well-known factor of enhanced immune response characterized with a higher peak level and longer half-life, and antibody dynamics after BNT162b2 vaccination is substantially affected by prior infection [28,29,30]. In individuals with prior infection, the protection against severe disease after second BNT162b2 dose may last considerably longer [28]. As shown in our study, the predictive power of prior COVID-19 status to maintain seropositive status 6 months after vaccination was very high, which theoretically may compensate for the waning antibody concentrations over time.

Despite a relatively high frequency of comorbidities reported in our HCW population, we did not find any significant association with antibody concentrations after adjustment for other confounding factors. In addition, age, sex, and BMI were not significant predictors of anti-SARS-CoV-2 concentrations 3 and 6 months after full vaccination with BNT162b2. There are conflicting findings on the relationship between age, sex, and underlying comorbidities and antibody levels after vaccination. While some studies reported lower post-vaccination antibody concentrations in males, elderly, and obese [31,32], others, in line with our study, yielded different results [33,34,35,36]. Moreover, the negative effect of underlying comorbidities may be strongly age-dependent. Discrepancies in estimates of humoral response between studies may be partially explained by differences in study protocols, population tested, statistical approach, and analytical performance of the assays. Indeed, the antibody magnitude is strongly assay- and timing-dependent [37].

Lack of adverse symptoms after vaccination and its correlation with antibody response are other controversial issues. A general assumption exists that lack of ARs does not necessarily mean that the vaccine is less efficient and may not provide sufficient protection. Indeed, the immune system is composed of different cell types and proteins where each element collaborates with others, complementing and strengthening its function to provide defense against infection. Our study clearly showed a significant and negatively correlated link between lack of symptoms after first and even more so after second dose of BNT162b2 vaccine and anti-RBD SARS-CoV-2 IgG in a large HCW cohort. Our findings contradict a previously published study which assessed whether adverse effects induced by first and second dose of BNT162b2 vaccine were associated with the magnitude of anti-SARS-CoV-2 IgG concentration. In their study, Coggins et al. also concluded that individuals with either low or high symptom scores had comparable levels of anti-SARS-CoV-2 RBD-specific IgG [14]. The disparities between our and this study are difficult to explain; however, our study cohort was larger, with measurements of antibodies performed 3 and 6 months after vaccination. Moreover, sampling and measurements were performed within a strict timeframe, in a single hospital center.

Our study had a few limitations which should be acknowledged. We did not evaluate the antibody response and its magnitude before and immediately after vaccination. Other components of immune protection, including T cells, may contribute to humoral response expressed by anti-SARS-CoV-2 levels. We did not measure neutralizing antibody status; however, studies have suggested a very high correlation with anti-SARS-CoV-2 IgG against RBD [38]. Our results are limited to predominantly female Caucasian HCWs as they constituted the most representative medical professional group in our hospital.

## 5. Conclusions

In conclusion, in this large cohort of HCWs, the immune antibody response after two doses of BNT162b2 vaccine was independently and remarkably associated with prior COVID-19 exposure, time since vaccination, and ARs after either dose of vaccine. Easily reportable ARs may facilitate the identification of the immune response in HCWs, especially in those who will require additional booster doses of vaccine to elicit their immune protection against novel SARS-CoV-2 variants.

## Figures and Tables

**Figure 1 vaccines-10-02125-f001:**
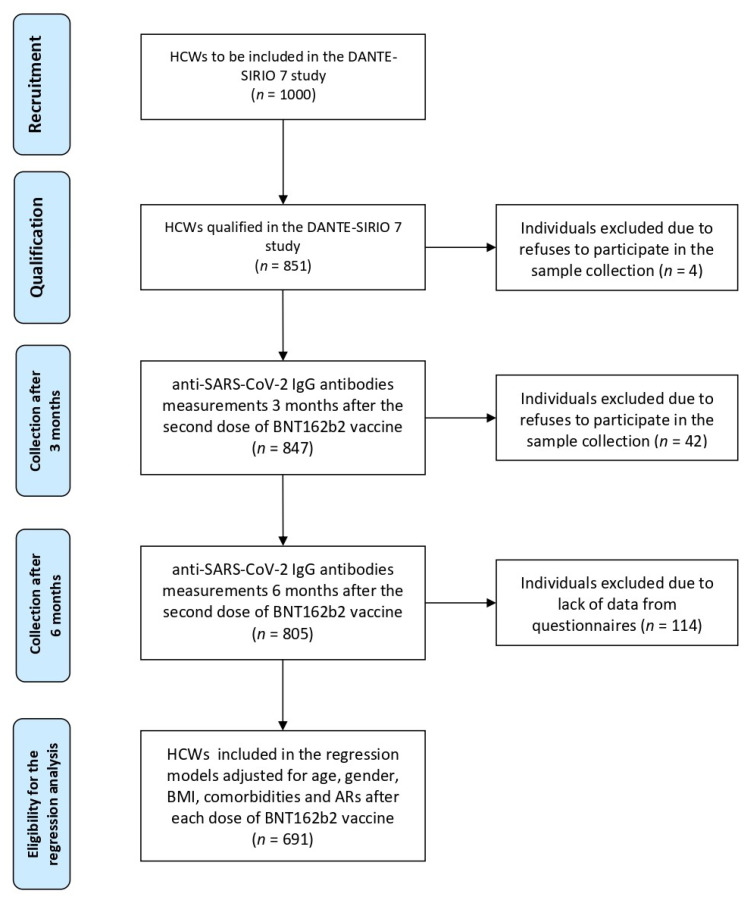
Flowchart for selection of the study participants. HCWs, healthcare workers.

**Figure 2 vaccines-10-02125-f002:**
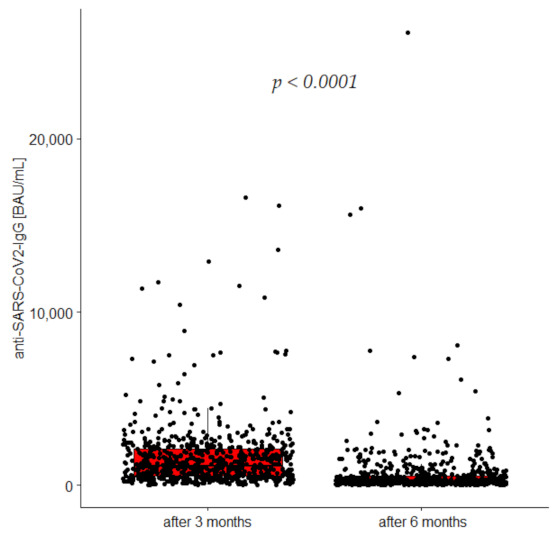
Anti-SARS-CoV-2 IgG concentrations 3 and 6 months after vaccination.

**Figure 3 vaccines-10-02125-f003:**
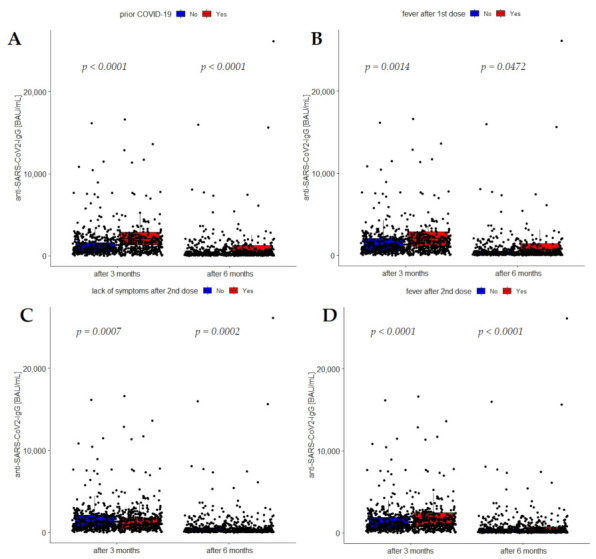
Anti-SARS-CoV-2 IgG concentrations 3 and 6 months after vaccination with regard to (**A**) prior-COVID-19 infection, (**B**) lack of symptoms after the second dose of the vaccine, (**C**) fever after the first dose of the vaccine, and (**D**) fever after the second dose of the vaccine.

**Figure 4 vaccines-10-02125-f004:**
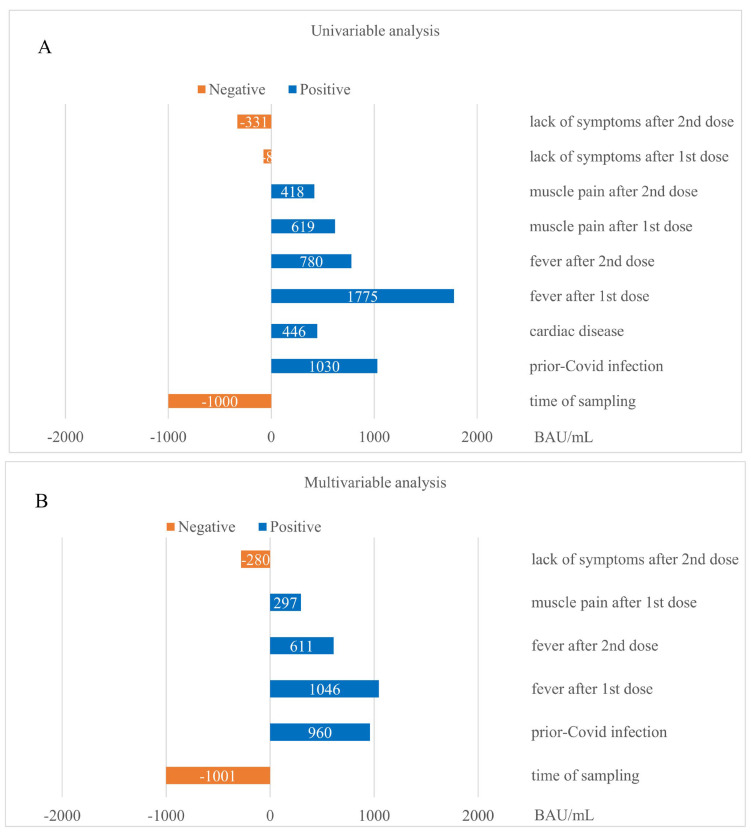
Quantitative impact of determinants on SARS-CoV-2 IgG concentrations, shown as an average absolute change in antibody levels expressed in BAU/mL for each statistically significant determinant: (**A**) in the univariable model, and (**B**) in the multivariable model.

**Table 1 vaccines-10-02125-t001:** Baseline characteristics of study participants.

Variable	All (*n* = 847)	Non-COVID-19 (*n* = 601)	Prior-COVID-19 (*n* = 246)	*p*
Age (years)	45 (34–53)	44 (32–53)	46 (37–53)	0.057
Gender [*n* (%)]	Female 677 (79.9%)	Female 485 (80.7%)	Female 192(78.4%)	0.001
	Male 170 (20.1%)	Male 116 (19.3%)	Male 54 (21.6%)	
BMI (kg/m^2^)	24.5 (21.9–28.1)	24.4 (21.8–27.5)	24.7 (22.4–29.0)	0.034
Overweight [*n* (%)]	265 (31.3%)	189 (31.4%)	76 (30.9%)	0.874
Obese [*n* (%)]	127 (15.0%)	83 (13.8%)	44 (17.9%)	0.131
Smoking [*n* (%)]	108 (12.7%)	77 (12.8%)	31 (12.4%)	0.933
Allergy [*n* (%)]	166 (19.6%)	127 (21.2%)	39 (12.8%)	0.079
Diabetes [*n* (%)]	55 (6.5%)	40 (6.7%)	12 (4.8%)	0.328
Hypertension [*n* (%)]	156 (18.4%)	96 (16.0%)	60 (24.4%)	0.004
Hyperlipidemia [*n* (%)]	146 (17.2%)	97 (16.2%)	49 (20.0%)	0.237
Cardiac diseases [*n* (%)]	61 (7.2%)	36 (6.0%)	25 (10%)	0.042
Thromboembolic diseases [*n* (%)]	26 (3.0%)	19 (3.1%)	7 (2.9%)	0.808
Autoimmune diseases [*n* (%)]	122 (14.4%)	81 (13.4%)	41 (16.7%)	0.230
CKD G4, G5 [*n* (%)]	9 (1.1%)	7 (1.1%)	2 (1.0%)	0.650
Pulmonary diseases [*n* (%)]	43 (5.1%)	31 (5.2%)	12 (4.8%)	0.866
Cancer [*n* (%)]	58 (6.8%)	43 (7.1%)	15 (6.2%)	0.580
SARS-CoV-2 IgG (BAU) after 3 months	1145 (543–2095)	829 (452–1640)	2042 (1267–2812)	0.0001
SARS-CoV-2 IgG (BAU) after 6 months	225 (100.0–510.3)	163.7 (86.5–329.4)	517.0 (251.2–1049.3)	0.0001

**Table 2 vaccines-10-02125-t002:** Comparison of the frequency of adverse reactions after the first and the second dose of BNT162b2.

Adverse Reactions	After First Dose of Vaccine [*n* (%)]	After Second Dose of Vaccine [*n* (%)]	*p*
None	142 (16.7%)	150 (17.7%)	0.265
Malaise	149 (17.6%)	215 (25.4%)	0.0001
Loss of smell	0	1 (0.1%)	NA
Loss of taste	2 (0.2%)	2 (0.2%)	1.00
Feverish state (<38 °C)	101 (11.9%)	159 (18.8%)	0.0001
Fever (>38 °C)	38 (4.5%)	118 (13.9%)	0.0001
Runny nose	5 (0.6%)	7 (0.8%)	0.772
Cough	6 (0.7%)	6 (0.7%)	NA
Sore throat	12 (1.4%)	15 (1.8%)	0.560
Dyspnea	4 (0.5%)	12 (1.4%)	0.079
Respiratory failure requiring oxygen therapy	1 (0.1%)	0	NA
Muscle pain	142 (16.8%)	237 (28.0%)	0.0001
Gastrointestinal complaints	16 (1.9%)	33 (3.9%)	0.014
Headache	107 (12.6%)	200 (23.6%)	0.0001
Injection site soreness	651 (76.9%)	548 (64.7%)	0.0001
Others	68 (8%)	102 (12%)	0.008

NA: not applicable.

## Data Availability

Not applicable.

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
