# Peer review of "Determinants of the Level of Anti-SARS-CoV-2 IgG ANTibodiEs after Vaccination (DANTE-SIRIO 7) Study in a Large Cohort of Healthcare Workers"

_vaccines, 2022, doi:10.3390/vaccines10122125_

Round 1

Reviewer 1 Report

The manuscript by Krintus and colleagues investigates various parameters following vaccination with two doses of the mRNA vaccine BNT162b2 vaccine that was given to a large group of healthcare workers (HCW). Using standard immunoassays, the authors measure anti-SARS-CoV-2 IgG responses (levels) against the receptor binding domain (RBD) at three- and six-months post-immunization.  The anti-SARS-CoV-2 responses were then correlated with parameters including a feverish state after the first and second doses of vaccine,  respiratory failure, muscle pain, headache injection site soreness, and GI complaints. Overall, I found the manuscript to be well-written and easy to follow.  The methods used were straightforward and the information presented in Tables 1 and 2, which included the baseline characteristics of the patients in the study and the adverse reactions that were followed, respectively,  were clear.  conclusions were backed by statistical analyses. The salient results of the study indicated that the presence of a fever after both vaccine doses, prior exposure to sars-CoV-2, and muscle pain after the first dose positively correlated with higher antibody titers.  These findings may be of value to the  SARS-CoV-2 immunization field.  My comments are mostly editorial in nature.

1.  Throughout the manuscript “Covid” is used. The disease was named by the WHO as CorOnaVirus Infectious Disease-2019.  COVID-19 should be used throughout the manuscript.

2. The following sentence (at the end of the Introduction), “Therefore, our study aimed to evaluate anti-SARS-CoV-2 IgG concentrations, after the full vaccination cycle with the BNT162b2 vaccine, during 6 months follow-up and then compare these results to the COVID-19 history and severity of symptoms during the dis-ease and after the first and second vaccine dose to identify major determinants of antibody response following vaccination,” should be broken into two sentences.

My suggestion:

Therefore, our study was aimed at evaluating anti-SARS-CoV-2 IgG concentrations, after two doses (the full vaccination cycle) with the BNT162b2 vaccine at 3- and 6-months post-vaccination. We compared these results with the COVID-19 history and severity of symptoms during the disease following vaccination to correlate major determinants of antibody response following vaccination.

3. Figure 1. In the upper two boxes to the right, “Individuals excluded due to refuses to participate in sample collection,” should be “Individuals excluded due to refusal to participate in sample collection.”

In the bottom box to the right, “Individuals excluded due to lacking data from questionnaires,” should be “Individuals excluded due to lack of data from questionnaires.”

4. Legend to Figure 4: “Quantitative impact of determinants on SARS-CoV-2 IgG concentrations, shown as an average absolute change in antibody levels expressed in BAU/mL for each determinant statistically significant: A. in the univariable model; B. in the multivariable model.

I suggest the following change (in bold):

“Quantitative impact of determinants on SARS-CoV-2 IgG concentrations, shown as an average absolute change in antibody levels expressed in BAU/mL for each statistically significant determinant: A. in the univariable model; B. in the multivariable model.

5. References section: The titles of some references are written in lower case while with others are written with the initial letter in upper case.

Author Response

Reviewer 1

  1. Throughout the manuscript “Covid” is used. The disease was named by the WHO as CorOnaVirus InfectiousDisease-2019.  COVID-19 should be used throughout the manuscript.

 We have corrected the text accordingly.

  1. The following sentence (at the end of the Introduction), “Therefore, our study aimed to evaluate anti-SARS-CoV-2 IgG concentrations, after the full vaccination cycle with the BNT162b2 vaccine, during 6 months follow-up and then compare these results to the COVID-19 history and severity of symptoms during the dis-ease and after the first and second vaccine dose to identify major determinants of antibody response following vaccination,” should be broken into two sentences.

My suggestion:

Therefore, our study was aimed at evaluating anti-SARS-CoV-2 IgG concentrations, after two doses (the full vaccination cycle) with the BNT162b2 vaccine at 3- and 6-months post-vaccination. We compared these results with the COVID-19 history and severity of symptoms during the disease following vaccination to correlate major determinants of antibody response following vaccination.

Thank you for this comment. We corrected the sentence to be more readible:

Therefore, our study was aimed at evaluating anti-SARS-CoV-2 IgG concentrations, after two doses (the full vaccination cycle) with the BNT162b2 vaccine at 3- and 6-months post-vaccination. We compared these results with the COVID-19 history and severity of symptoms during the disease and following vaccination to correlate major determinants of antibody response after vaccination

  1. Figure 1. In the upper two boxes to the right, “Individuals excluded due to refuses to participate in sample collection,” should be “Individuals excluded due to refusalto participate in sample collection.”

In the bottom box to the right, “Individuals excluded due to lacking data from questionnaires,” should be “Individuals excluded due to lack of data from questionnaires.”

 It has been corrected as suggested.

  1. Legend to Figure 4: “Quantitative impact of determinants on SARS-CoV-2 IgG concentrations, shown as an average absolute change in antibody levels expressed in BAU/mL for each determinant statistically significant: A. in the univariable model; B. in the multivariable model.

I suggest the following change (in bold):

“Quantitative impact of determinants on SARS-CoV-2 IgG concentrations, shown as an average absolute change in antibody levels expressed in BAU/mL for each statistically significant determinant: A. in the univariable model; B. in the multivariable model.

It has been changed as suggested. 

  1. References section: The titles of some references are written in lower case while with others are written with the initial letter in upper case.

We wrote references according to the citation proposal copied from pubmed.

Reviewer 2 Report

In this manuscript, Krintus et al., analyzed the anti-SARS-CoV-2 IgG level after BNT162b2 vaccination. They analyzed the data with large cohort of HCWs and found that the anti-SARS-CoV-2 response after BNT162b2 vaccine was independently associated with prior-Covid exposure, time since vaccination and the occurrence of symptoms. Overall, the experiments and analysis are well performed. I have a few minor comments for this manuscript.

Minor

1.     Figure 3: It’s hard to understand what blue and red square means. Please add the explanation in the legend.

2.     Figure 3: Is there significant difference of IgG level between blue-red and post-3-6-month vaccination? Please add the statics analysis data.

3.     Do the authors know/speculate with which variants the donors were infected?

4.     Typo: The word “BNT162b2” changed to “BN16b2” in some sentence. Please fix them.

Author Response

In this manuscript, Krintus et al., analyzed the anti-SARS-CoV-2 IgG level after BNT162b2 vaccination. They analyzed the data with large cohort of HCWs and found that the anti-SARS-CoV-2 response after BNT162b2 vaccine was independently associated with prior-Covid exposure, time since vaccination and the occurrence of symptoms. Overall, the experiments and analysis are well performed. I have a few minor comments for this manuscript.

Minor

  1. Figure 3: It’s hard to understand what blue and red square means. Please add the explanation in the legend.

In accordance with the reviewer's request, we have added this information to both Figures 2 and 3.

  1. Figure 3: Is there significant difference of IgG level between blue-red and post-3-6-month vaccination? Please add the statics analysis data.

We have added the p value to the Figure 2 and Figure 3, accordingly.

  1. Do the authors know/speculate with which variants the donors were infected?

In accordance with the referee's suggestion, we have added to the revised manuscript the following sentence:

The emergence of a highly transmissible and more pathogenic delta variant (B.1.617.2) of SARS-CoV-2 in Poland [9] seems to be responsible for breakthrough infections observed in our study.

  1. Typo: The word “BNT162b2” changed to “BN16b2” in some sentence. Please fix them.

We are sorry for this omission. We have corrected the word for BNT162b2 in the entire manuscript.